# A Review of Pectin-Based Material for Applications in Water Treatment

**DOI:** 10.3390/ma16062207

**Published:** 2023-03-09

**Authors:** Javier Martínez-Sabando, Francesco Coin, Jorge H. Melillo, Silvia Goyanes, Silvina Cerveny

**Affiliations:** 1Centro de Física de Materiales (CSIC, UPV/EHU)-Materials Physics Center (MPC), Paseo Manuel de Lardizabal 5, 20018 San Sebastián, Spain; 2Donostia International Physics Center (DIPC), 20018 San Sebastián, Spain; 3Laboratorio de Polímeros y Materiales Compuestos (LP&MC), Departamento de Física, Facultad de Ciencias Exactas y Naturales, Universidad de Buenos Aires, Buenos Aires 1113, Argentina; 4Instituto de Física de Buenos Aires (IFIBA), CONICET—Universidad de Buenos Aires, Ciudad Universitaria (C1428EGA), Buenos Aires 1113, Argentina

**Keywords:** pectin, low methoxylation degree, adsorption, cross-linking, structural models, wastewater, remediation, pollution

## Abstract

Climate change and water are inseparably connected. Extreme weather events cause water to become more scarce, polluted, and erratic than ever. Therefore, we urgently need to develop solutions to reduce water contamination. This review intends to demonstrate that pectin-based materials are an excellent route to detect and mitigate pollutants from water, with several benefits. Pectin is a biodegradable polymer, extractable from vegetables, and contains several hydroxyl and carboxyl groups that can easily interact with the contaminant ions. In addition, pectin-based materials can be prepared in different forms (films, hydrogels, or beads) and cross-linked with several agents to change their molecular structure. Consequently, the pectin-based adsorbents can be tuned to remove diverse pollutants. Here, we will summarize the existing water remediation technologies highlighting adsorption as the ideal method. Then, the focus will be on the chemical structure of pectin and, from a historical perspective, on its structure after applying different cross-linking methods. Finally, we will review the application of pectin as an adsorbent of water pollutants considering the pectin of low degree methoxylation.

## 1. Introduction

Water is at the basis of social and economic development; it is vital to every human being; it is essential for growing food, health, and safety. Moreover, water is related to the economic vitality of our society, and, at present, there is no substitute for water on the earth’s crust. Water has a Sustainable Development Goal (SDG) that is dedicated exclusively to its promotion (Goal 6, “Ensure availability and sustainable management of water and sanitation for all”), and it has a crucial role in a good number of the SDGs while forming an essential factor in others. Nevertheless, due to the effects of climate change, which are being felt right now [1], water is becoming a more scarce and less predictably available resource in many world regions [2]. Water scarcity is among the principal problems that many societies and the entire world will face during the XXI century, and it is already affecting every continent.

Water on the planet is under pressure because of population growth (we need more and more water to produce food), the overexploitation of freshwater resources used in different industries, and climate change. As the population grows and the environment is affected by climate change, access to drinking water decreases. More than 1300 children under five die every day because of diseases caused by unsafe drinking water, poor sanitation, and hygiene. The global demand for freshwater will rise by 40% in 2030, putting pressure on water-stressed areas. On top of this, the energy demand will go up by 50%, further stressing water as 90% of all power generation is water-demanding. The world is heading toward a global water crisis, and it is urgently necessary to secure freshwater for all.

Water pollution is the release of substances into water bodies (oceans, seas, lakes, rivers, streams, canals, etc.) that make water unsafe for human or aquatic ecosystems’ use. Nowadays, contamination is observed in almost all sources of available water bodies. Agriculture products such as fertilizers, manure, pesticides, sediment, pharmaceuticals, and household products (detergents or soaps) are the primary drivers of extremely diffuse pollution into rivers, streams, and estuaries. This results in low-oxygen eutrophic ‘dead’ zones [3] that are spreading worldwide. In addition, metal pollution, mostly from mining and technology manufacturing, is another growing threat, especially in developing countries that lack environmental regulations and adequate wastewater treatment. Plastic waste, especially micro-plastics, is another growing concern, with the most significant footprint existing in the industrial sectors of Asia [4].

A fundamental environmental challenge posed by water contamination is the presence of emerging contaminants (such as medicines, pharmaceutical products, industrial chemicals, heavy metals and metalloids, and pesticides), which reduce drinking water quality drastically. These emerging contaminants are gaining notable prominence in water remediation research because they possess an intricate molecular nature and are very hard to detect and remove (see Figure 1 for a list of compounds). Over the last two decades, pharmaceuticals [5] (including personal care products) and heavy metals have been considered substantial contaminants in the environment as their presence in water has grown substantially. We need to consider, for instance, that the EU takes second place for pharmaceutical sales for human and veterinary consumption on a global scale. In addition, there is an increasing trend in their consumption. Consequently, there are several reports on the contamination of aquatic environments by pharmaceuticals in Europe’s water bodies. Some examples are the feminization of fish [6,7] or high antibiotic levels in the water [8].

Pharmaceuticals can enter the environment by different routes [9], including agricultural runoff and urban or industrial waste. Medicines used for humans are mostly discharged from wastewater treatment plants (WWTP) [9], whereas veterinary antibiotics discharged in water occur via excretion. This is because antibiotics are only partly metabolized and might retain their antibiotic activity. Between 80 and 90% of the amounts ingested are excreted and released into the environment, contaminating water and soil [10]. Consequently, bacterial resistance increases the risk of toxicity towards aquatic species or humans in the case of the ingestion of this water [11]. Therefore, a mixture of antibiotics and their metabolites travel through the sewage system to the WWTP, where their complete elimination is impossible. Thus, antibiotics can reach natural aquatic systems (surface waters, soils, and sewage sludge). The presence of antibiotic and antifungal pharmaceuticals may play a role in accelerating the growth and spread of resistant bacteria and fungi with the consequent risk to human health. It is also necessary to note that no clear association has been demonstrated between pharmaceuticals in the environment and their direct impact on human health. However, the World Health Organization (WHO) has reported on the possible effects of long-term exposure on vulnerable populations [12]. More research is still needed to comprehend and evaluate pharmaceutical products concerning their environmental concentrations and levels of risk [13].

On the other hand, reports on heavy metals such as lead, zinc, arsenic, cadmium, mercury, chromium, or nickel are also disturbing. Although between 2010 and 2020 and due to legislation [14], the releases of heavy metals from industries declined significantly in Europe’s water bodies, we cannot observe the same reduction in other continents such as America or Asia. In addition, independent of industrial wastes, for instance, arsenic is naturally (i.e., geogenic) present in rocks and can accumulate in different aquifers [15].

In contrast to medicines, heavy metals are found intrinsically in the earth’s crust and reach the surface environment through some geological events (for instance, volcanic eruption) or due to anthropogenic activities such as smelting, mining, or pesticide application. Moreover, technology-based activities (for example, industries, including color pigments or alloys) have also generated the presence of heavy metals in water bodies. Through leaching, infiltration, and runoff, metals and metalloids are transported to surface and ground waters. Once released into the aquatic environment, they are usually adsorbed into sediment particles but can be re-released during storms or other climatic events. Upon heavy metals are in water bodies, they do not experience chemical degradation similar to other pollutants but prevail in the water for a long time. The presence of heavy metals in water can introduce health issues [16,17] that include the risk of cancer and diabetes, skin lesions, renal injuries, cardiovascular disorders, neurotoxicity, and neuronal damage, among others.

All these risks related to water already have profound social and financial impacts. Regarding financial consequences, they are not reflected in day-to-day business costs. The global economic systems treat water as an unlimited and infinite resource with little value, leading to widespread waste and misuse. Regarding social impacts, we can mention the relocation of high populations of people due to pollution with the induced impoverishment of resettles, illnesses, and the increment in the gap between urban and rural areas, including social conflicts among different stakeholders.

For all these reasons, we need to develop solutions to maintain unpolluted water urgently. Here, we will review the use of pectin as an adsorbent because it is a promising alternative for water remediation. Figure 2 reproduces the number of publications in Clarivate Analytics’s Web of Science from 1970 to the present using the key “pectin and adsorption”. These findings show a substantial and consistent increase, indicating that pectin has recently sparked a significant amount of interest in the scientific community for its potential application as an absorbent.

In the following, we will summarize the water remediation technologies emphasizing adsorption as the preferred method. Then, the focus will be on the chemical structure of pectin, highlighting how source and extraction methods affect its final properties, such as the degree of methoxylation. Then, we will discuss previous works relating to the insolubilization of pectin (i.e., cross-linking methods) and its structural models developed from a historical perspective. Finally, we will examine the application of pectin as an adsorbent of water pollutants such as metal ions and dyes [18,19] considering the low degree of methoxylation pectin.

## 2. Water Remediation Technologies

Commonly, multiple water treatments are needed to purify raw water before it can be distributed. The treatment applied depends on the type of pollutant in the water. In each region of the planet, the contaminants differ. For instance, in North and South America, we could find high concentrations of arsenic, but this is not the case in oriental Africa. In addition, different regulations [20] are applied (distinct in each country, region, or continent) to meet the quality criteria of raw water input. Therefore, the water treatments applied are different in different world regions. In the following, we will summarize some of the existing water treatments and, after that, focus on adsorption techniques as a preferred method of water remediation.

The most common pollutants in wastewater (presented worldwide) are microbes, bacteria, or suspended solids that can be removed using traditional methods. However, other contaminants (pharmaceuticals or industrial chemicals) are difficult to remove, transform, or degrade. Among the conventional methods, we can mention filtration (water passes through a column of bed and bank materials to remove particulate material and debris from the raw water, in addition, microorganisms and algae can also be removed using slow sand filters) or aeration (to transfer oxygen into water and remove gases and volatile compounds by air stripping). In addition, phase separation (such as sedimentation, i.e., the separation of suspended solids (such as algae) by gravity) is also generally used. We can also mention chemical processes (such as oxidation involving the transfer of electrons from a reduced reagent to the reduced chemical species), which aim to transform putrescible contaminants into innocuous products and biological processes. Finally, biological treatments that rely on bacteria and other tiny organisms using cellular processes are used to break down organic waste. This treatment aspires to create a system where decomposition results can be easily collected for proper disposal.

All these methods mentioned above are typically used in wastewater treatment plants (WWTPs), and they play a crucial role in environmental preservation and are essential for modern urban life [21]. However, the design of WWTPs is based on the need to reduce organic and suspended solids, and they are not intended to remove emerging contaminants [22]. These types of contaminants (for instance, pharmaceuticals, heavy metals, or industrial chemicals) are difficult to remove, transform, or degrade [23]. Remediation efficiencies in WWTPs can be less than 10% in the case of pharmaceuticals [24], and its removal has rarely been considered an objective, but this must now be of primary concern when water is used, for instance, in agriculture or livestock. It is important to remark that wastewater contains emerging contaminants, and after the water passes through the WWTP, it still maintains several of these contaminants [25]. The presence of pharmaceuticals in freshwater has been reported in several research investigations [26,27,28]. Therefore, as most WWTPs are not equipped to deal with these products, WWPTs are new sources of pharmaceuticals for the environment [29,30].

For these reasons above, over the last twenty years, several new treatments for cleaning water have been developed with the primary goal of including these emerging contaminants. Among the most recent developments for water treatment, we can mention the bio-electrochemical systems in which a reactor simultaneously performs water treatment and energy production [31,32]. These systems consist of a chamber divided into two parts by an ion exchange membrane. They can also be employed for organic matter removal [33], desalination [34], or metal recovery [35]. However, these systems produce low energy [36] with a high fabrication cost [37] and are difficult to employ in large-scale facilities [38].

Another relatively new treatment is ultraviolet irradiation technology (UV). It is generally used as a disinfection process because UV radiation, in the wavelength range from 250 to 270 nm, has a germicidal effect [39]. In this case, water flows around a series of UV lamps, and therefore, it can be easily implemented into traditional WWTPs. Mercury lamps have been used as UV radiation sources, but more recently, ultraviolet light-emitting diodes have become available with many advantages over mercury lamps [40]. In addition, advanced oxidation processes via semiconductor photo-catalysis water treatment systems [41,42,43] are suitable for use in water and wastewater treatment facilities. They can treat industrial wastewater polluted with high loads of organic substances or metals. However, this technology is still costly and not massively used in WWTPs.

On the other hand, since the 1990s, membrane filtration technology has been developed in surface water treatment [44,45]. Membrane filtration is a pressure-driven technology with pore sizes ranging from nm to microns. Membrane filtration involving reverse osmosis (RO), ultrafiltration (UF), microfiltration (MF), and nanofiltration (NF) in drinking water production has increased rapidly over the past decade [46,47,48]. However, the main problem of membrane technology is its high fouling tendency [49] (deposition of matter in the membrane, which dramatically affects its performance).

Adsorption is also a promising and low-cost water purification technology [50]. Although it is an old technique, it has recently been used for numerous purposes. [51,52,53]. We favor this technology, among the other mentioned methods, mainly because of the process’s simplicity, cost-effectiveness, energy requirements, and reversibility [54,55]. After the absorbent is used, it is possible to desorb the pollutants and regenerate the adsorbent for later reuse [56]. Therefore, adsorption is a technique to recover chemicals that contribute to the circular economy [57,58]. Additionally, compared to other approaches, the materials needed to create adsorbents are typically less expensive, and the technology does not require electricity to be employed effectively [59].

Adsorption occurs when molecules in a liquid bind to the surface of a solid substance. The adsorption process establishes weak chemical bonds (ion exchange) and physical forces (hydrogen bonds, van der Waals, electrostatic interactions) between the adsorbent surface and the adsorbate. It depends on diverse variables such as contact time, pH, temperature, surface-volume ratio, and concentration of pollutants [60,61]. Adsorbents with large surface areas and high porosity usually show high adsorption efficiencies. Different adsorbent materials for water remediation, such as activated carbon, have been known since early 1900 [62]. Table 1 shows a partial list of examples for adsorbents used in wastewater treatment. Some of them are carbon materials such as graphene oxide, bio-char [10,63], or activated carbons [64,65,66]), silica-based materials [67,68,69], inorganic materials such as zeolites [70,71,72], mineral clays [73,74], and organic materials such as polysaccharides [75,76]. In the latter case, the way in which the removal of contaminants occurs is not sufficiently clear in the literature. As we will see below in this review, it could be an ion interchange or a combination of ionic interchanges plus other types of interaction.

The adsorption process is typically studied by adsorption kinetics [88], isotherms [89], and by applying models. Kinetics represents the amount of the pollutant adsorbed by the adsorbent as a function of the time with an initially fixed pollutant concentration in the water. Pseudo-first order (PFO), pseudo-second-order (PSO), Elovich, and Boyd’s external and internal diffusion models are the most used adsorption kinetic models [88]. The isotherm is represented as the adsorptive capacity of the material measured as a function of the initial concentration at equilibrium, i.e., when the kinetics has reached the plateau. Langmuir, Freundlich, Sips, Henderson, Temkin, and Redlich Peterson are the most used models to explain adsorption [89]. All these models have been previously reviewed in the literature (see, for instance, references [88,89]). Therefore, we will not review this topic here.

## 3. Pectin—Chemical Structure

Pectin is an anionic heteropolysaccharide that is present in vegetable cell walls. Pectin is particularly predominant in fruit peels, especially in citrus peel and apple pomace, but also in passion fruit rind, pomelo, and banana peel [90]. The pectin backbone chain comprises α-(1,4)-linked D-galacturonic acid (GalA) residues [91] (mainly homogalacturonan (HG), rhamnogalacturonan I (RG-I), rhamnogalacturonan (RG-II)) linked by α 1-4 glycosidic bond. The homogalacturonan domain (HG) is the most abundant and linear domain (around 60%). Additionally, as shown in Figure 3a, pectin has different branch domains, such as xylogalacturonan with sugars (xylose, apiose, rhamnose, galactose, arabinose, fructose, among others).

The carbon 6 of the D-galacturonic acid residues can be methyl-esterified (COOCH_3_) or carboxylated (COO^−^, in its deprotonated form), as indicated in Figure 3b. It also can be acetylated at the O-2 and/or O-3 position of the GalA residues, although this is less common [91]. Depending on the ratio of the methyl-esterification of these residues, pectin shows a different esterification degree (DE).

Carboxyl groups are hydrophilic and can coordinate with metal ions, whereas methyl-esterified are phobic. Therefore, by maximizing the number of COO^−^ (i.e., low degree of esterification), the adsorption efficiency against metal ions is much higher than pectin with a high amount of COOCH_3_ groups.

DE is a magnitude that is related to the gelling mechanism of the pectin, and it classifies pectin into two groups: high methoxyl pectins (HM), where more than 50% of the galacturonic acid residues are esterified, and low methoxyl (LM) pectins, where less than 50% of the galacturonic acid residues are methoxylated. Independent of the DE, pectin is a water-soluble biopolymer. Therefore, it should become insoluble to be used as an adsorbent in aqueous environments. This can be achieved by cross-linking the pectin with various cross-linking agents, as described in the next section.

Pectin sources (i.e., the plant used to extract pectin) strongly influence the galacturonic acid content and the degree of methoxylation obtained after extraction [92,93]. These two variables directly affect pectin’s ability to form a gel. The most common sources of pectin are apple pomace and citrus peels. Whereas the galacturonic acid content for apple pomace is between 20 and 44%, and the orange peel has been higher in comparison.

There are different extraction methods to isolate pectin from vegetables, which also affect the structure of the pectin, and it involves several steps (pretreatment, extraction, purification, concentration, precipitation, drying, and standardization) [94]. The preferable method is to mix the cell-wall materials in hot or cold acidified water [95,96] because the highest extraction yield is achieved. It is also possible to add chelating agents [97] such as cyclohexane-diamine-tetra acetic acid (CDTA), oxalate, or ethylene-diamine-tetra acetic acid (EDTA) to promote the release of pectin. On the other hand, if instead an acidic alkaline extraction is used, the length of the galacturonic acid, the methoxylation degree (DM), and acetylation (DA) decrease [98].

The possible uses of pectin in different industries are abundant and of different natures, mainly in the health and pharmacy sector, food applications, and packaging. As pectin is a natural component of vegetables and fruits, it is an exceptionally safe multifunctional food additive (E440) that is used, for example, as a texturizer or gelling agent. In addition, pectin is popular in several scientific fields because of its availability, safety, relatively low price, and functionality. Specific structural attributes developed by pectin’s functional groups or the attachment of chemicals on the molecule make it a good candidate for several purposes, such as food innovations [99], nutritional remediation [100], drug delivery [101], illness treatment [102], tissue engineering [103], and other approaches [104]. Notably, pectin is resistant to gastrointestinal hydrolyzing enzymes and acidic/alkaline media, which favors its application in colon delivery via an oral route under a specific condition.

In particular, pectin, with a low level of esterification, also has a high application value and broad application prospects as a functional food ingredient [105]. It also has been studied as a forthcoming biomaterial for tissue engineering and biomedical applications [106]. Pectin films have also been used in packaging, although there are some limitations because of their inadequate mechanical properties and high hydrophilicity of the pectin films [107,108]. This issue can be solved by blending with other polymers [109] or incorporating cross-linkers and/or filler material [110].

## 4. Insoluble Pectin—Cross-Linking Agents That Generate Gelation

As mentioned above, pectin is a water-soluble material. Therefore, it must be insolubilized to be used as an adsorbent to remove pollutants from water. Different ionic crosslinking agents were used for pectin, with the cations that were monovalent (Na^+^, K^+^), divalent (Ca^2+^, Cu^2+^, Sr^2+^ Ni^2+^ Zn^2+^ Cd^2+^ Pb^2+^ or Mg^2+^), or trivalent (Al^3+^, La^3+^, and Fe^3+^) being most used. Pectin can also be cross-linked using non-ionic cross-linkers such as glutaraldehyde [111] or laccase [112]. However, ionic crosslinking is the most widely used and suitable method because it has the principal advantage of releasing contaminants (in some cases, recovering a priceless pollutant) and recovering the absorbent that can be reused.

Among all the possible ions, the divalent cations are the most commonly used elements for LM pectin gelation. After cross-linking (independent of the cross-linking agent), pectin undergoes gelation, producing a three-dimensional network in the form of xerogels (vacuum drying), hydrogels (wet gels), aerogels (supercritical CO_2_ drying), and cryo-gels (freeze-drying) depending on the drying conditions [113]. The primary characteristic of wet gels is their ability to bind considerable quantities of water, thus increasing their volume. In addition, the swelling favors the access of pollutants to pectin.

The gelation mechanism of pectin is mainly governed by the methyl-esterification degree. Therefore, the gel formation mechanism differs for high-methoxyl (HM) and low-methoxyl (LM) pectins [114]. Apart from the methyl-esterification degree, the gelling process is influenced by ionic strength, molecular weight, and pH [115]. In addition, after cross-linking, the final structure of pectin depends on the cross-linking cation used.

There are fewer reports on LM pectin cross-linking using monovalent cations such as sodium (Na^+^) and potassium (K^+^). Cross-linking with these monovalent cations is produced at a low pH (approximately from pH = 2 to pH = 4) [116,117,118,119]. The cross-linking with monovalent ions diminishes the repulsive charges between pectin chains and promotes chain-chain association via hydrogen bonding [116,119]. It was suggested that at pH = 4.5, the cross-linking would more effective using K^+^ than Na^+^ because of the charge screening, galacturonic acid de-esterification, hydrogen bond changes, and electronic attraction [116]. Moreover, by increasing the pH to higher values (alkaline conditions), it was also found [116] that Na^+^ ions generated much stronger gels than those induced by K^+^ in HM pectin. In accordance, Yoo and coworkers [120] also studied poly (methyl esterase) mediated de-esterified citrus pectin when cross-linked with Na^+^, K^+^, and Li^+^. In agreement with Chen et al. [116], they observed that stronger cross-linking was produced using Na^+^ followed by K^+^ at pH = 7, whereas the reverse was true for pH = 5. A similar study was performed by Strom et al. [119,120], where the influence of L^+^, Na^+^, and K^+^ on LM pectin rheology in an acid solution was evaluated. These investigations revealed that K^+^ formed the strongest gels, followed by Na^+^ and Li^+^. Furthermore, LM pectin was cross-linked by NaCl (0.05M) with and without ZnCl_2_ [118] and could form nanoparticles. Li^+^ does not form gels at any of the tested conditions. This behavior was ascribed to an increase in the ionic radius of the hydrated cation.

Trivalent cations have also been employed as cross-linking agents for pectin [121,122,123,124]. It was found [121,122] that independent of the methyl-esterification degree, Al^3+^ binds pectin chains had the condition of pH ≥ 4. In a rheological investigation [122], Ca^2+^, Cu^2+^, Al^3+^, and La^3+^ ions were used as cross-linking agents. All these cations can form pectin gels. It was stated that the weakest gel was formed with La^3+^, followed by Ca^2+^, similar to Al^3+^, while Cu^2+^ was one order of magnitude more substantial than the other cations. Finally, it was reported that the trivalent cations Fe^3+^, Ce^3+^, Pr^3+^, and Nd^3+^ could establish cross-links in pectin [124]. In that work, the formation of isolated rhamnogalacturonan II dimers was reported. Still, there were no reports regarding what type of structure the pectin chains formed in the presence of these trivalent metals.

Apart from monovalent or trivalent cations, divalent cations such as Ca^2+^ are the most used cross-linking agents for pectin. The gelation of LM pectin with calcium ions occurs by forming junction zones according to the so-called “egg-box” model, which has been initially described for alginates [125]. The lower the pectin esterification degree, the more divalent the cation binding sites that are available on pectin, and the ability to absorb divalent cations increases. The following section will discuss the egg-box model from a historical perspective.

Many works have proven the effective cross-linking of LM pectin using Ca^2+^ [126,127,128,129,130,131,132,133,134]. In addition to Ca^2+^, the cross-linking of LM pectin with Cd^2+^ and Cu^2+^ was reported for drug delivery applications [135]; Pectin was also cross-linked with Sr^2+^ and Zn^2+^ to form aerogels for diclofenac sodium controlled release [136]. Pectin cross-linked by Cu^2+^ ions was reported to be used as a scaffold for gold nanoparticles [137]. Moreover, pectin was cross-linked with a 2% (*w*/*v*) ZnCl_2_ solution [138] to evaluate the potential healthcare applicability, and Das and coworkers [139] used LM pectin cross-linked with Zn^2+^ to encapsulate a colon-specific drug delivery microsphere.

On the other hand, other molecules have been used to induce the gelation of LM pectin. Yoshimura and coworkers [111] described the cross-linking of the LM pectin with ethylene glycol diglycidyl ether and glutaraldehyde or Ca^2+^. They found that glutaraldehyde produced pectin gels, whereas the ethylene glycol diglycidyl ether did not show an apparent gel formation. Ullah and coworkers [140] reported the cross-link of pectin with methylene bisacrylamide and ammonium persulfate as initiators. Pre-saponified pectin was also cross-linked with adipic acid [141] or adipic acid dihydrazide, which was previously involved in the oxidation of pectin [142]. Chen and coworkers compared the cross-link using adipic acid dihydrazide and the classic Ca^2+^ cross-link, reporting that the adipic acid dihydrazide cross-link significantly improved cell adhesion. Another example of a non-ionic gelling agent is laccase, which was employed to cross-link sugar beet pectin in the work, as performed by Jung and Wicker [143], where they noticed an increase in the molecular weight of pectin by chromatographic techniques.

McCuet et al. [144] electrospun LM pectin with poly(ethylene oxide) and later cross-linked either with Ca^2+^ or oligochitosan. They found that the morphology of the resulting fibers was preferable to cross-linking them with oligochitosan, and it was more suitable for tissue engineering as it has no apparent cytotoxicity due to a positive surface charge. Finally, Mongkolkitikul et al. [145] cross-linked citrus pectin with citric acid and FeCl_2_ to encapsulate ibuprofen. They discovered that the diffusion coefficient was significantly improved using FeCl_2_ as a cross-linker.

In conclusion, we described that the effective cross-link of the LM pectin could be achieved through several cross-linking agents of different natures. Divalent cations are the most widely used among these agents for relevant industrial or commercial applications. Therefore, in the next section, we describe the structure of the LM pectin cross-linked by divalent metal cations.

## 5. Structural Models of Cross-Linked LM Pectin and Alginates

This section will discuss the structural models proposed for pectin after cross-linking. From a historical perspective, we will mainly focus on the interaction of pectin with divalent ions because this is the most relevant cross-linking agent for obtaining a material suitable for water remediation.

### 5.1. Historical Perspective of Structural Models Induced by Calcium

In 1973, Grant and coworkers [125] first proposed the interaction between a polysaccharide chain (such as alginate or pectin) and divalent cations in terms of the egg-box model. Initially, this model introduced alginates: a non-branched polysaccharide composed of two monosaccharides, α–L guluronic and β-D mannuronic acids [146]. Using circular dichroism, the authors suggested a cooperative inter-chain mechanism of ion binding involving two chains associated with the coulomb interaction between the active sites and the metal ion. More specifically, two opposite α-L-guluronate sequences paired together to form a structure with cavities, within which Ca^2+^ ions were accommodated through specific coordination interactions between two free carboxyl groups.

Five years later, Morris et al. [147] studied alginate chain gelation by X-ray diffraction coupled with circular dichroism using different calcium concentrations. Their results showed that the primary inter-chain association mechanism was regularly dimerizing alginate or poly (guluronate) chain segments. This structure results in cross-linked dimers with a geometric design that resembles an egg box (see Figure 4). Compared with the first proposal of the “egg-box” model, the dimers can further aggregate laterally into multimers, as revealed by small-angle X-ray scattering. In addition, the sugar ring belonging to guluronic acid and the polymer chain adopts a characteristic zigzag shape. Studying the lateral aggregation of dimers is essential to understanding the properties of these biomaterials in terms of their structure to explain what happens during a water remediation process. On the other hand, knowing the correct features and operating principle of the “egg-box” model helps to understand the possible pollutants of adsorption models from wastewater.

In 1981, Walkinshaw and Arnott [148] added, for the first time, the concept of the junction zone for the “egg-box” model for poly(galacturonic) acid and a high methoxylation degree of calcium pectate (i.e., pectin cross-linked with metal ions). In this new model, for poly(galacturonic) acid, chain–chain interactions were stabilized by intermolecular hydrogen bonds formed between several adjacent D-galacturonic acid units and hydrophobic bonding between methyl esters. Similarly, hydrophobic binding from segments of methyl groups and specific intermolecular hydrogen bonds stabilized calcium pectate. The main interactions between pairs of chains could be the bridges formed by calcium ions, which incorporate into their coordination shells two-polyanion oxygen atoms from one chain and three from the other. In addition, in 1982, Powell et al. [149] investigated rhamnose distribution for the formation of stable poly(galacturonic) acid inter-chain junctions for high and low degrees of methyl(esterification). Their results confirm the Walkinshaw and Arnott hypothesis [148]. They also indicate that the length of the poly(galacturonic) acid sequences between rhamnose interruptions (i.e., hairy regions) is approximately constant (i.e., they prove the existence of the junction zone).

Later, Kohn [150] studied the interaction between carboxyl groups and divalent cations in pectin fragments, resulting in a complex formation. The interaction exhibits different affinities towards divalent cations: Ca^2+^, Sr^2+^, and Zn^2+^ ions are bound according to the well-known “egg-box” model, whereas Cd^2+^, Cu^2+^, and Pb^2+^ form intra- and inter-molecular complexes. This work proved that the “egg-box” structure only formed with specific metal ions. This revolutionized the egg-box model because, depending on the concentration and interaction with metal ions, LM-pectin could be cross-linked in different ways with different properties.

At this point, and to bring all the above studies together, a book entitled the chemistry and technology of pectin [151] was first published in 1991. This event changed scientists’ attitudes toward pectin research because of the exponential growth of publications in the following years.

During the first ten years of the 2000s, scientists began to study the “egg-box” model using various approaches, and different reconsiderations of the model have been proposed. A molecular simulation study [152], made on pairs of galacturonate oligomers, found a shift along the chain axis in the association of associated galacturonate chains. Therefore, they proposed the “shifted egg-box” model to explain the gelation mechanism of galacturonate chains, where two shifted antiparallel pectin chains are necessary to form the egg-box dimer.

In 2007, the “shifted egg-box” model for Ca-alginate gels was re-examined [153] and confirmed using X-ray diffraction measurements. Unlike previous “egg-box” models developed for pectin, in this case, the authors considered the helical conformation of the main chain, which had not been considered before. These results suggest that instead of a 2/1 helical conformation, as previously proposed for the egg-box model [154], a 3/1 helical conformation for the junction zones was energetically more favorable for these materials. In the same year, Fang et al. [155] studied the Ca^2+^-alginate gel of high (long polymer chain) and low molecular weight (short polymer chain) using isothermal titration calorimetry. They proposed a three-step binding behavior of calcium to alginate to form “egg-box” dimers: (i) the interaction of Ca^2+^ with a single guluronate unit forming mono-complexes (i.e., a carboxylic group with Ca^2+^); (ii) the propagation and formation of egg-box dimers via the pairing of these mono-complexes; and (iii) the lateral association of the egg-box dimers, generating multimers. The short chains are quite rigid, and inter-cluster association is the only possible way for the “egg-box” dimers to aggregate laterally. This aggregation results in an increase in molecular size. However, long chains are more flexible and have smaller clusters. The intra-cluster association, in this case, results in a reduction in molecular size (see Figure 5).

In 2008, Fang and coworkers [156] compared pectin with alginate for calcium-binding behavior. LM pectin is similar to alginate, but a two-step mechanism is involved. Whereas step I can be attributed to a mono-complex between Ca^2+^ ions and a polygalacturonate chain, step II is the formation of egg-box dimers through the pairing of the mono-complexes. In addition to steps I and II, the binding of alginate with Ca^2+^ includes a third step that is assigned to the lateral association of egg-box dimers.

In 2010, Gohil [157] studied the structural reorganization of pectin and alginate films after calcium binding by X-Ray diffraction and dynamic mechanical measurements. His results suggest that the structural reorganization of molecular network structure, after binding with Ca^2+^, destroyed some existing pectin crystalline tie points, resulting in its amorphization, as described by the “fringe-micellar” structure. In the same year, Fraeye and coworkers [130] summarized in detail how the characteristics of the final pectin gel are affected by different “intrinsic” and “extrinsic” parameters such as the amount and distribution of methyl ester, chain length, pectin concentration, amidation, and acetylation degrees, molecular weight, calcium content, temperature, and pH. Table 2 highlights how these parameters influence the final characteristics of the gel.

Ventura and coworkers [158] published a SAXS analysis of Ca^2+^–LM pectin gels with different calcium concentrations, coupled with molecular dynamics simulation studies. They proposed a new “egg-box” model considering semi-flexible and no linear chains. This chain flexibility was not considered in any of the previous models developed. They showed [158] that rod-like and point-like cross-links between neighboring pectin molecules could occur. With the increased Ca^2+^ concentration, the number of road-like cross-links decreased while the number of point-like cross-links increased. In this study, the authors proposed that the cross-linking scheme was mainly governed by the branched nature of pectin, as opposed to the linear nature of alginate.

Wang et al. [159] studied the high and low methoxyl pectin self-assembly molecules, which were regulated by calcium ions using atomic force microscopy (AFM) in the following years. The addition of calcium ions increased the viscosity of the low-methoxyl pectin solution. Otherwise, the viscosity of the high-methoxyl pectin solution remained stable. AFM confirmed that the esterification degree and calcium concentration cause different binding methods with calcium ions between pectin governed by the branched nature of pectin [158]. The formation of dimers by lateral aggregation was observed only for the LM-pectin. However, for the HM-pectin, the large number of esterified galacturonic acid residues limits pectin fiber aggregation due to the nonspecific hydrophobic interaction and hydrogen bonds. This study was another confirmation of the two-step binding behavior of calcium to pectin. The AFM study showed that there were two possible ways of dimer aggregation. In accordance with the Walkinshaw and Arnott hypothesis [148], the lateral aggregation of dimers involved hydrogen bonds between hydroxyl groups of different pectin fibers. However, dimers can aggregate along the pectin fiber, as the picture shows.

Finally, we wanted to indicate that by looking at Figure 3, Figure 4, Figure 5 and Figure 6, it is possible to understand how the “egg-box” model has evolved since it was proposed in 1973.

### 5.2. Recent Evolution of Structural Models for Pectin Is Cross-Linked with Other Metals

As mentioned above, pectin is susceptible to cross-links with other cations apart from calcium. Specifically, in 2015, Assifaoui et al. [160] published the structural differences found in LM pectin when cross-linked with Ca^2+^ and Zn^2+^. These results show that Ca^2+^ cations only interact with carboxyl groups and form more homogeneous pectin network fibers (i.e., the well-known “egg-box” dimers). In contrast, Zn^2+^ also interacts with hydroxyl groups, resulting in a less homogeneous cross-linked pectin network (see Figure 7).

In 2016, Huynh et al. [161] investigated the binding mechanism using different cationic metals (Zn^2+^, Ca^2+^, Ba^2+^, Mg^2+^) by isothermal titration calorimetry, viscosity measurements, and molecular dynamics simulations in poly (galacturonic) acid. In this framework, a monodentate interaction meant the interaction between a carboxyl group of galacturonic acid and a cation. In contrast, a bidentate interaction means the binding of two oxygen atoms of the galacturonic acid (from two different pectin chains) and a cation. They reported that the interaction between divalent cations and poly (galacturonic) acid is monodentate for Mg^2+^ and Zn^2+^ and bidentate for Ba^2+^ and Ca^2+^. Moreover, the binding mechanism for the divalent cations, Zn^2+^, Ca^2+^, and Ba^2+^, can be associated with mono-complexation and point-like cross-links or related to the appearance of the dimmers formation, depending on the molar ratio between the divalent cation and galacturonic acid (R = Metal^2+^/Gal). This conclusion was based on the number of water molecules coordinated with the different cations. In the case of Zn^2+^, Ca^2+^, and Ba^2+^, the coordination with water was weaker in strength than those of Mg^2+^. During the cross-link, the metal ions lose one water molecule for the Zn^2+^ and two for Ca^2+^ and Ba^2+^, respectively. However, Mg^2+^ strongly interacts with water and remains weakly bound to poly(galacturonic) acid by sharing water molecules from its coordination shell with the carboxylate groups. Finally, two years later, Huynh et al. [162] studied the gelation kinetics of pectin induced from divalent cations (Zn^2+^, Ca^2+^, Ba^2+^, Mg^2+^) by viscoelastic and turbidity measurements. Their results confirmed that the cations’ diffusion kinetics was lower for Zn^2+^ than for Ca^2+^ and Ba^2+^. However, for the Mg^2+^, the gel was not formed, supporting a previous study [161].

To finish this section, Table 3 summarizes the milestones in developing the structural models of pectin when cross-linked with metal cations.

## 6. Properties of Pectin That Influence the Absorption of Metal Ions

Several investigations focused on using pectin as an adsorbent to remediate heavy metals from the water. It is important to remark that we do not review the values of adsorption capacities for each pectin type developed in the literature. This task has been conducted in other revisions, such as references [19,76]. Instead, we compared the adsorption capacities of the LM pectin since, as explained above, we believe this material is more appropriate for water remediation purposes. Table 4 shows an overview of the maximum binding capacities for different divalent metals for pectin with an LM degree (<50).

Cataldo et al. [163] prepared pectate and poly(galacturonate) calcium gel beads for mercury (II) removal. Based on a different pH value study, they found that the best pH range for Hg^2+^ removal was between 3 and 3.6. Celus et al. [164] studied the ability of citrus pectin to adsorb Fe^2+^, emphasizing two structural properties: the degree of methyl esterification (DM) and the degree of blockiness (ratio of non-methyl esterified GalA units present in blocks to the total amount of GalA units) of citrus pectin. They found that the DM and DBabs influenced pectin-Fe^2+^ interactions: the higher the DBabs or lower DM, the higher the Fe^2+^ binding capacity.

In another study, pectin micro-gel particles were also used to remove methylene blue (MB) [165]. The authors found very high absorption ratios (see Table 4), even with time shorter than 20 min. Additionally, LM pectin was used to clean mineral soils polluted with Cu^2+^ [166]. Recently, a sweet potato residue modified by high hydrostatic pressure (HHP)-assisted pectinase was prepared [167]. This material was used for Pb^2+^ removal. They showed that the modified sweet potato pectin exhibited more excellent adsorption performances for Pb^2+^ and Cu^2+^ than natural pectin.

LM Pectin was also mixed with other biopolymers, such as chitosan, to prepare chitosan-pectin gel beads, which were synthesized via a green method [168] to remove a collection of heavy metals (Cu^2+^, Cd^2+^, Hg^2+^, and Pb^2+^). They analyzed different variables such as the effect of pH, contact time, heavy metals concentration, temperature, and adsorption mechanism. They found that adding pectin increased the adsorption capacities, porosity, and stability of the adsorbents. The infrared analysis of the adsorbents indicates that the interaction between heavy metals and chitosan-pectin gel beads is due to the complexation with functional groups such as carboxyl, hydroxyl, amine, and amide.

Mata et al. [169] prepared sugar-beet pectin xerogels using residues of the sugar industry. Their study is in conditions of continuous biosorption and not on stationary conditions as the above presented. In this case, the adsorbent was used for metal recovery from effluents in continuous systems. They studied different experimental conditions: feed flow rate and bed height (amount of biosorbent) and found that the best conditions for Cu^2+^ sorption in column reactors were: 3 g of biomass, 25 mg/L metal, 2 mL/min feed flow rate, and a reverse feeding system.

It is necessary to remark that there was a substantial dispersion of results obtained in the literature in relation to the adsorption capacity of pectin-based adsorbents. This is because the adsorption capacity of pectin strongly depended on its chemical characteristics and relied on both the source on which pectin was obtained and the extraction method. For instance, an early work by Kartel et al. [175] showed that apple pectin had the highest affinity for Co^2+^ ions, whereas, in the case of beet pectin, the affinity was better for Cu^2+^ and Cd^2+^ ions. In contrast, citrus pectin showed a different preference for Ni^2+^, Zn^2+^, and Pb^2+^ ions. These different affinities depend on the structural differences of pectin. For instance, the galacturonic acid content for apple pomace is between 20 and 44% [176], whereas, for orange peel, it has been between 66 and 70% on a dried basis. In addition, apple pectin obtained from the different extraction procedures was highly methylated (from 54.5% to 79.5%) [98], but the methoxylation degree was much lower for orange peels. Thus, the source of pectin and the way it is extracted determines the structural properties of pectin, which, in turn, defines the adsorption capacity of heavy metals.

Among the different structural properties of pectin, the more critical parameter related to heavy metals binding is the methoxylation degree (DM) [91,177]. When DM is small, more carboxyl groups are available on pectin, which can interact with the metal ion, increasing the adsorption capacity. In addition, it has been reported that the pattern of methyl esterification is also crucial for determining adsorption capacities [178].

The influence of DM on the adsorption of metal ions has been studied for the adsorption of Zn^2+^ [179], Fe^3+^ [164], Ca^2+^, Zn^2+^, Fe^2+^, and Mg^2+^ [180]. The influence on other structural parameters of pectin, such as the degree of acetylation (DA), chain length, or the branched domains (for instance, RG I, RG II, etc., see Figure 2a), have been less studied concerning the adsorption capacity.

The molecular weight of pectin also influenced the adsorption capacity. The lower the molecular weight, the higher the adsorption capacity [181]. This effect occurs because small pectin chains can reveal additional active sites that can result in a more significant electrostatic attraction to capture more heavy metal ions [172]. In addition, a longer pectin chain can form a firmer gel in which the metal ions cannot penetrate.

In addition, due to the high degree of heterogeneity regarding chemical composition and physicochemical properties such as molecular weight, the degree of esterification, dispersion, and galacturonic acid content of the pectin, predictions of the metal-binding capacity of different pectin batches are quite tricky.

## 7. Future Perspectives

Pectin-based adsorbents have very high capacities for the adsorption of heavy metals. However, the adsorption of heavy metal ions through ion exchange depends on the interaction between the ion used for cross-linking and the heavy metal ion. This removal method is, therefore, perfect for any pollutant in ion form but not for other types of contaminants, such as bacteria or pharmaceuticals. This is one of the topics on which basic science needs to implement improvements to impose pectin-based absorbents on the market.

Related to this above, the egg-box model of pectin gelation by calcium has been widely studied since it was proposed in 1973. The formation of this egg-box structure is crucial to the adsorption capacity of pectin because the pollutant capture is produced because of an interchange between the contaminant and the calcium ion. However, some research gaps still exist in the formation of the egg-box structure. For instance, it is still debated which structure is formed by the cross-link with other cations in addition to Ca^2+^.

On the other hand, as mentioned, Ca^2+^ allows the formation of the “egg-box” structure with rod-like cross-linking forming dimers, which can aggregate each other. In the case of Zn^2+^, the cross-link is point-like. It results in a final different structure that affects the final properties of the pectin. However, a network formed by other divalent metals has not been elucidated yet. In the case of trivalent metals, some of them cannot cross-link pectin, and it is still unclear why. Concerning trivalent cations that effectively cross-link LM pectin means, there is still a gap in terms of which mechanism they follow and what final structure they form between the metal and the biopolymer.

It is still unclear which are the best sources and parameters for pectin extraction and synthesis because they play an essential role in the final intrinsic properties of the biopolymer, such as ramification, active sites, and steric hindrance that can affect interactions with the multivalent metal ions. Standardized protocols are needed to obtain suitable pectin-based water remediation materials consistently.

Another variable that still needs to be developed for pectin-based adsorbents is the type of pectin’s pore structure, which is formed when cross-linked with different agents. This variable could be crucial to improve the adsorption capacities of pectin. Thus, optimizing the water diffusion through the hydrogel means that it is easier for active sites and adsorbates to come into contact, increasing the adsorption capacity of the hydrogel against heavy metals.

An additional improvement aspect is its potential capacity to remediate pollutants that are different from heavy metals such as pharmaceuticals. This can be conducted by modifying the pectin chemical structure and generating new functional groups that are susceptible to the adsorption of contaminants, adding other fillers or functionalized nanoparticles, or altering the cross-linking agent. Therefore, more basic-oriented research is needed to obtain the simple, cheap, and large-scale production of pectin base material to remove multiple pollutants from aqueous media.

Finally, large-scale industrial applications still need to be practiced to clean pharmaceuticals and other micro-pollutants from water using pectin based-adsorbents. Further investigations are, therefore, needed to fulfill a large-scale extraction from by-products of the juice industry, for example, to further stimulate the circular economy.

## 8. Summary

This review shows that pectin-based materials have enormous potential for specific pharmaceutical, food, industrial, and biomedical applications. In particular, we discussed the development of membranes to absorb pollutants from liquid water. Pectin has the advantage that it is considered a widely available and cheap bioresource.

Considering heavy metal adsorptions, pectin-based materials are exciting for the development of water remediation adsorbents because adsorbed heavy metals can be desorbed and re-covered for future uses, changing the pH in a controlled environment. When heavy metals are desorbed, pectin hydrogels can be used for several adsorption–desorption cycles. Once the lifecycle of the hydrogel is accomplished, it can be discarded with no environmental impact, as if it is biodegradable once, pollutants can be desorbed. However, the adsorption capacities depend on several factors, as discussed here; therefore, more fundamental research is encouraged in this field.

In particular, we demonstrated that LM pectin is a suitable material to remediate water from heavy metals and an exceptional candidate for investigation as a multipollutant remediator following different cross-linking strategies and extraction methods.

## Figures and Tables

**Figure 1 materials-16-02207-f001:**
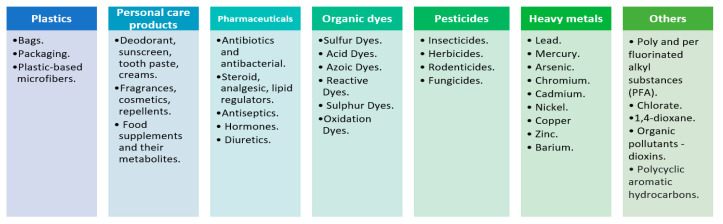
Common examples of emerging contaminants in water.

**Figure 2 materials-16-02207-f002:**
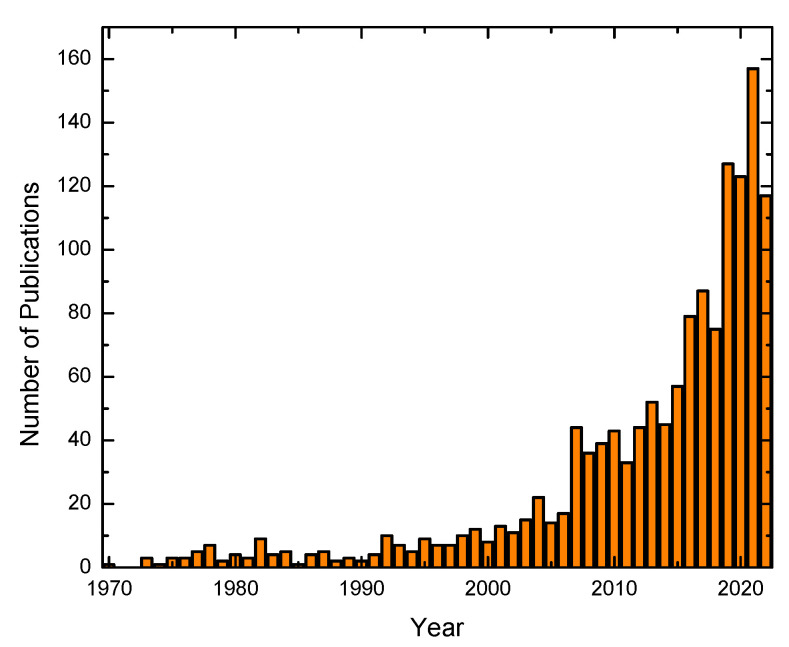
Publications in pectin water remediation for 1970–2022. Data are the number of hits of a search for publication titles containing the keywords “pectin” and “adsorption” using the Web of Science.

**Figure 3 materials-16-02207-f003:**
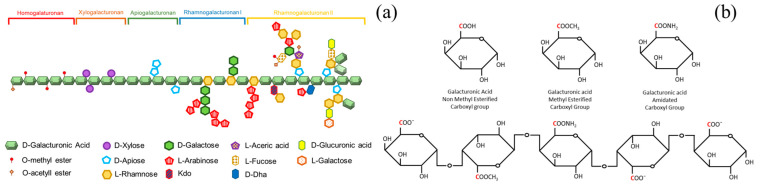
(**a**) Representation of pectin structure. Pectin has different polysaccharides (homogalacturonan, xylogalacturonan, apiogalacturonan, rhamnogalacturonan II, and rhamnogalacturonan I, as shown in the figure). (**b**) The schematic representation of the poly(galacturonic) acid chain indicates the Carbon 6 and galacturonic acid units with different functional groups at Carbon 6.

**Figure 4 materials-16-02207-f004:**
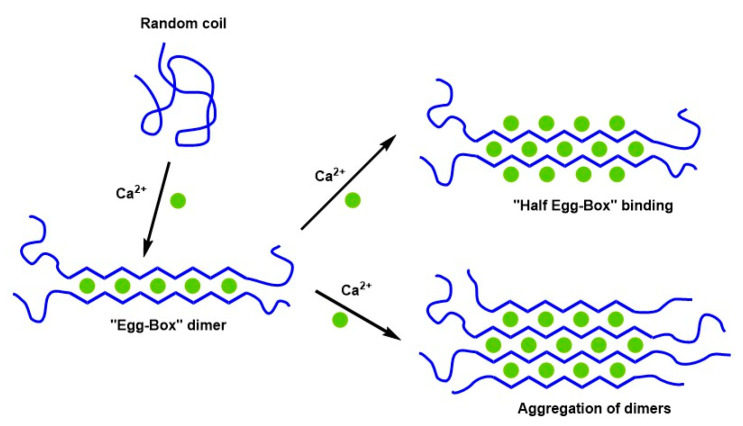
Schematic representation of calcium cross-linking in pectin or alginate [147].

**Figure 5 materials-16-02207-f005:**
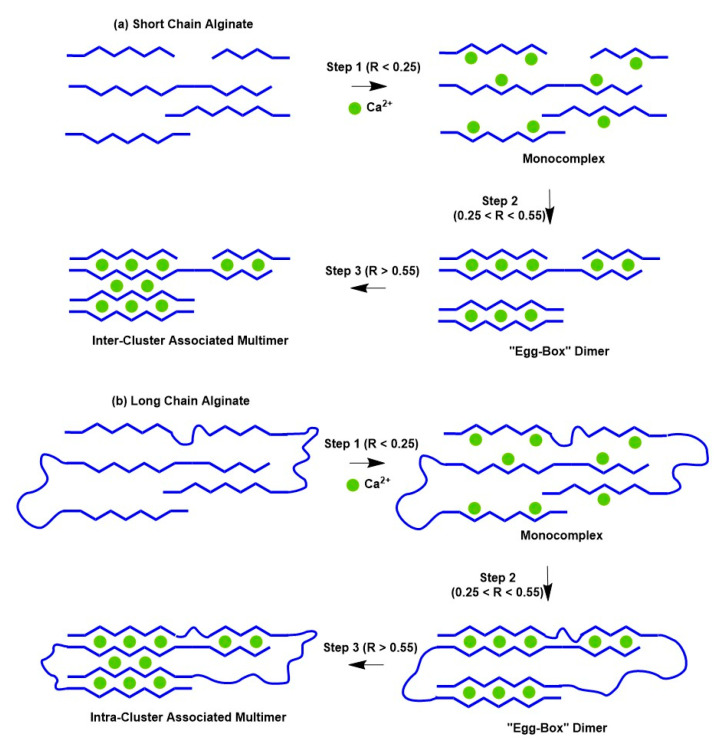
Ca^2+^ binding to alginate in multi-steps is shown schematically for the (**a**) Short-chain alginate and (**b**) Long-chain alginate, respectively. R stands for the Ca^2+^ to the guluronate residue feeding ratio [119].

**Figure 6 materials-16-02207-f006:**
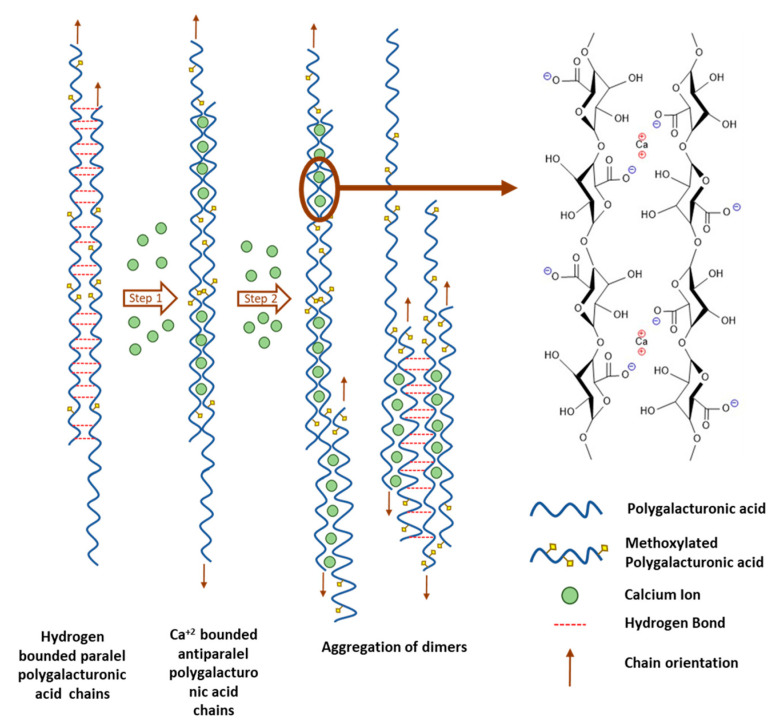
Cartoon representing the evolution of Ca^2+^ cross-linking of LM pectin and its final structure [159].

**Figure 7 materials-16-02207-f007:**
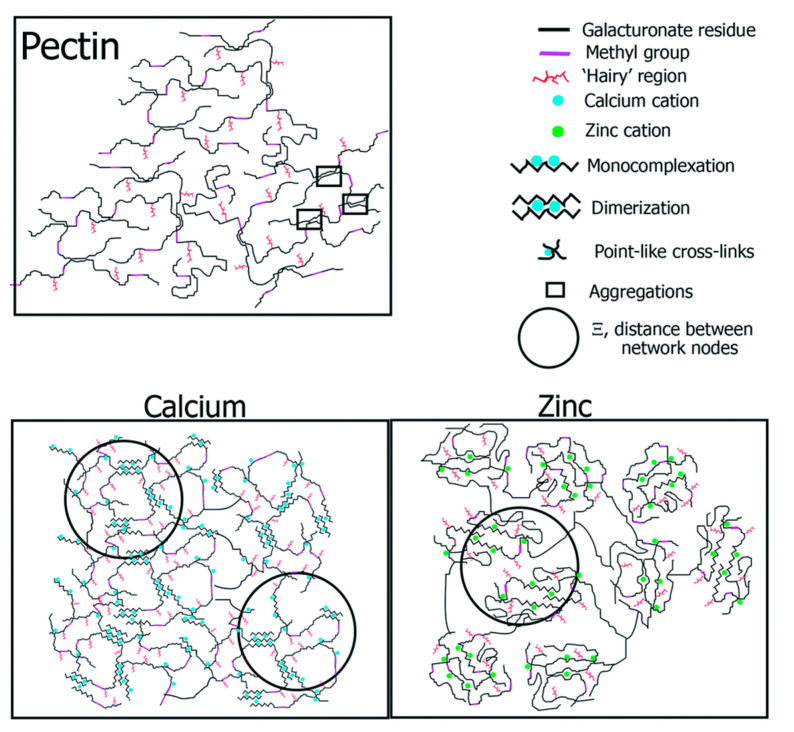
Suggested network scheme for calcium and zinc pectate complexes at R = 0 (pectin alone) and R = 0.44. Figure extracted from reference [160].

**Table 1 materials-16-02207-t001:** Adsorbents used for different pollutants (list restricted to papers from 2010).

Adsorbent Type	Contaminant	Reference
Activated carbon	Dyes, heavy metals	[64,65,66]
Graphene oxide	Pharmaceuticals, Organic compounds, metal ions	[77,78]
Silica-based materials	inorganic and organic pollutants	[67,68,69]
Zeolites	Petroleum, fluoride, nitrate, dyes, heavy metals, cesium	[70,71,72]
Biochar	Heavy metals	[10,79,80]
GO-biochar	Persulfate, metal ions, dyes, pharmaceuticals	[63]
Mineral Clays (Montmorillonite, Bentonite, Kaolinite, clinoptilolite, etc.)	Nuclear waste, pharmaceuticals	[73,74]
Sugar beet pulp	Nitrites and nitrates	[81]
Organic polymer resin	Cationic and anionic surfactants, perfluoroalkyl acids	[82,83]
Poly(saccharide)-based materials	heavy metals (arsenic)	[75,76]
Poly (vinyl alcohol) nanofibers + Iron NPs	Arsenic	[84,85]
Poly (vinyl alcohol) nanofibers + L-cysteine	Arsenic	[56]
Polybutylene adipate terephthalate (PBAT) nanofibrous	Dyes, pharmaceuticals	[86]
Biohybrid membrane of polymeric nanofibers and free-living bacteria	Chromium (Cr-VI)	[87]

**Table 2 materials-16-02207-t002:** Parameters (intrinsic to pectin and extrinsic) that influence the gel formation.

Intrinsic Parameters	
Methoxylation degree (DM)	Decreasing DM: the number of sequences of non-methoxylated GalA residues long enough for the egg-box formation to increase. This results in a substantial increase in the calcium ion binding capacity.
Pattern of methoxylation	Pectin with a block-wise distribution of non-methoxylated carboxyl groups can associate with egg-box formation at a higher DM. Pectin with carboxyl groups randomly distributed does not form egg-box patterns.
Chain length	The lower the molecular mass of pectin, the lower the gel strength.
Branching	Large side chains are likely to cause steric hindrance, which may hamper pectin-pectin interactions.
Amidation	Amidated pectin can form stronger gels, especially at a low pH, because of the formation of hydrogen bonds between amide groups.
Acetylation	Acetyl groups drastically decrease the binding strength of pectin with calcium ions.
**Extrinsic Parameters (Environment)**	
Calcium content	The influence of calcium ion concentration on the gelation of LM pectin is mostly described in terms of a stoichiometric ratio: R = 2[Ca^2+^]/[COO^−^]. R = 0.5, theoretically, all calcium ions are bound, forming the “egg-box” structure.
Pectin content	When R is kept constant, gel strength increases with the polymer concentration.
pH	To form ionic cross-links between the pectin carboxyl groups and calcium ions, pectin needs to be charged, i.e., the carboxyl groups need to be dissociated. Above pH = 4.5, the gel properties are relatively independent of pH, but when the pH of a pectin-calcium gel decreases below 4.5, the charge density of pectin decreases and, consequently, its affinity for calcium ions decreases. However, this effect is partly compensated by forming hydrogen bonds between protonated carboxyl groups.
Temperature	At high temperatures, a chain scission is promoted, and, as a consequence, dimers are formed. These junction zones are stabilized upon cooling through hydrogen bonding, which is accompanied by cooperative calcium immobilization.

**Table 3 materials-16-02207-t003:** Historical development of the binding mechanisms between divalent cations and both poly(galacturonic) chains or pectin.

Year	Event	Reference
1973	First postulation of egg-box model for pectin	[125]
1978	Dimers formation	[147]
1981	Junction zones model	[148]
1982	Junction zone distribution	[149]
1987	Divalent cations coordination	[150]
1991	First book in pectin chemistry and technology	[151]
2001	Shifted egg box model for pectin gels	[152]
2007	Multi-step binding ions behavior	[153]
		[155]
2008	Alginate and pectin comparison	[156]
2010	Structural reorganization of pectin and alginate after calcium binding	[157]
2013	Egg-box model with semi-flexible chains	[158]
2015	Cross-link differences between different multivalent cations	[160]
2016		[161]
2018		[162]
2020	Different ways in pectin dimers aggregation	[159]

**Table 4 materials-16-02207-t004:** Summary of pectin-based adsorbents’ maximum adsorption capacity (q_max_) for different divalent metals. Here, we only considered pectin of a low methoxylation degree. The table indicates the pectin origin, type of adsorbent, cross-link agent (CA), degree of methylation (DM), and type of pollutant.

Origin	Adsorbent	CA	DM [%]	Pollutant	q_max_ [mg/g]	Ref.
Citrus fruit	Pectin	Ca^2+^	<50	Hg^2+^	1.7 mmol/g	[163]
Commercial pectin	Demethylated pectin	Ca^2+^/Mg^2+^	13	Fe^2+^	0.523 mol Fe^2+^/mol GalA	[164]
Commercial pectin	Pectin microgel particles	Ca^2+^	16.4	MB *	284 mg/g	[165]
Citrus peel	Pectin with Oxisol	Nr.	6.7	Cu^2+^	33.8 mmol/kg	[166]
	Pectin with Utisol			Cu^2+^	37.0 mmol/kg	
Sweet potato residue	HHP-AP ** modified pectin	Nr.	16.11	Pb^2+^	263.15 mg/g	[167]
	Pectin		28.01		163.93 mg/g	
Commercial pectin	Chitosan-pectin gel beads	Alkaline solution	6.68	Cu^2+^	169 mg/g	[168]
				Cd^2+^	177.6 mg/g	
				Hg^2+^	208.5 mg/g	
				Pb^2+^	266.5 mg/g	
Sugar beet pectin	Pectin xerogel beds	Ca^2+^	<50	Cd^2+^	0.151 mmol/g	[169]
				Pb^2+^	0.290 mmol/g	
				Cu^2+^	0.343 mmol/g	
Citrus pectin	Pectin microspheres	Ca^2+^	47.9	Pb^2+^	127 mg/g	[170]
	LM pectin microspheres		18.04		292 mg/g	
	Pectic acid microspheres		0.9		325 mg/g	
Citrus peel	Pectin and guar gum beds	Ca^2+^	20	Pb^2+^	104.8 mg/g	[171]
Citrus peel	Pectin-alginate beds	Ca^2+^	<50	Cu^2+^	2.79 µmol/bead	[163]
				Cd^2+^	3.4 µmol/bead	
Phyllospadix iwatensis	Native Pectin	Nr.	6.91	Pb^2+^	2.447 mM/g	[172]
	Hydrolized Pectin		2.54	Cd^2+^	1.643 mM/g	
				Pb^2+^	2.818 mM/g	
				Cd^2+^	2.396 mM/g	
Grapefruit peel	Biochar-pectin-alginate beads	Ca^2+^	17.5	Cu^2+^	80.6 mg/g	[173]
Commercial pectin	Pectin	Ca^2+^	<50	MB **	354.6 mg/g	[127]
Orange Waste	Pectin/cellulose microfibers beds	Ca^2+^	<50	Fe^2+^	98.0 mg/g	[174]
				Cu^2+^	88.5 mg/g	
				Cd^2+^	192.3 mg/g	

* MB: methylene blue; ** HHP-AP: high hydrostatic pressure assisted pectinase, Nr.: not reported.

## Data Availability

No new data were created in this study. Data sharing is not applicable to this article.

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
