# Peer review of "A Review of Pectin-Based Material for Applications in Water Treatment"

_materials, 2023, doi:10.3390/ma16062207_

Round 1

Reviewer 1 Report

This review paper provides a practical overview on application of pectin-based material for treatment of wastewater. Some important physical/chemical properties of pectin have been introduced and potential mechanisms involved in interaction between pectin molecules and pollutants have well described. The paper carries many critical aspects of the subject and can contribute to the current knowledge of the issue. However, there are some challenging shortcomings that should be addressed in the revised version of the paper as follows:

1. Authors may rethink about the title if ignoring comment 6.

2. Abstract is not self-representative! Half of the abstract are some basics that can be replaced with more critical points.

3. Keywords must be improved. Please use more representative ones, such as wastewater instead of water, etc.

4. Figure 2 is really necessary? Do you access all references published in the world?

5. Section 2 is a leading part but to some extent misleading. Please rearrange it such that readers can reasonably be directed to the necessity of your work.

6. Sections 3 to 6 are the core of your work. However, with respect to the title you have picked for your valuable work, they don’t cover the practical findings in the field of wastewater treatment. Esteemed authors are suggested to prepare a section before section 3 to review and present those works used pectin to treat wastewater. Preparing a table including the main technical results will be very useful.

7. Please break section 7 into sections “7. Future perspectives” and “8. Summary”. Then, in section 7 you can discuss more about the present scientific and technical gaps and potential future works to cover them.

Good luck,

Author Response

We thank the reviewer for taking the time to read our manuscript, and we thank the helpful comment for improving our review. We agree with most of their words and, therefore, we have modified our article in the following points.

  1. Authors may rethink about the title if ignoring comment 6.

As the reviewer suggested, we added a new section (see comment 6). Therefore, we leave the title as in the first version.

  1. Abstract is not self-representative! Half of the abstract are some basics that can be replaced with more critical points.

As the reviewer suggested, we re-write the abstract indicating more critical points.

  1. Keywords must be improved. Please use more representative ones, such as wastewater instead of water, etc.

A new list of keywords was provided.

  1. Figure 2 is really necessary? Do you access all references published in the world?

We think the Web of Science is a reliable source of scientific publications. Therefore, it is a usual way to indicate how the evolution related to indexed journals in this topic. We prefer to maintain this plot in the new version of the manuscript. On the other hand, we are not interested in the exact number of publications but in the tendency of publications in this field. A higher exponential tendency means higher interest in the topic.

  1. Section 2 is a leading part but to some extent misleading. Please rearrange it such that readers can reasonably be directed to the necessity of your work.

We are not sure what changes the reviewer wants us to apply. In any case, we rewrite section 2. We hope the reviewer is pleased with our changes.

  1. Sections 3 to 6 are the core of your work. However, with respect to the title you have picked for your valuable work, they don’t cover the practical findings in the field of wastewater treatment. Esteemed authors are suggested to prepare a section before section 3 to review and present those works used pectin to treat wastewater. Preparing a table including the main technical results will be very useful.

As the reviewer suggested, we have added more information about pectin when it is used as an adsorbent of pollutants in water. However, instead of adding a section before "3", we have completed this information in section 6. We have added a table including the main technical results, such as the degree of methoxylation and the adsorption capacity (see new table 4) of pectin based-materials, but only considering LM pectin adsorbents. We hope the referee finds this section appropriate. 

  1. Please break section 7 into sections “7. Future perspectives” and “8. Summary”. Then, in section 7 you can discuss more about the present scientific and technical gaps and potential future works to cover them.

We have broken section 7 as the referee suggested. In the new version of the manuscript, we have two new sections, “8. Future perspectives” and “9. Summary”.

Reviewer 2 Report

This manuscript reviews that pectin-based hydrogels are a good way to detect and mitigate contaminants in water. Twenty kinds of water remediation techniques are summarized and adsorption is emphasized as the ideal method. The chemical structure of pectin and pectin structures has been studied from a historical perspective by different cross-linking methods. The application of pectin as adsorbent for water pollutants with low methoxylation degree was also demonstrated. However, some points of the manuscript should be improved. Specific comments are given below.

1.    In Figure 2, lines 140 to 149 appear garbled.

2.    In the part of 2. Water remediation technologies, it is suggested that the author cite several literatures in the traditional method here.

3.    Line 282-285: “Specific structural attributes were developed by pectin's functional groups or attachment of chemicals on the molecule make it a good candidate for several purposes, such as food innovations, nutritional remediation, drug delivery, illness treatment, tissue engineering, and other approaches.” This part suggests that authors cite literature.

4.    Line 393: “5. Structural models of cross-linked LM pectin and alginates”. In this part, the author enumerates some examples in history and recent years, but does not give the proper differences.

5.    Line 553:6. Properties of pectin that influence the absorption of metal ions” the author can add more discussion in this section.

6.    Please carefully check the text for writing and grammar problems.

Author Response

We thank the reviewer for our manuscript's careful reading and helpful comments. In the following, we respond to all your comments and the changes performed in the manuscript.

  1. In Figure 2, lines 140 to 149 appear garbled.

In the new version of the manuscript, we have changed the colors used to prepare Figure 2. We think the figure is correct in the latest version of the manuscript.

  1. In the part of  Water remediation technologies, it is suggested that the author cite several literatures in the traditional method here.

We extend the references corresponding to section 2. In the new version of the manuscript, we have added references (see further references 20 to 87). In addition, we have added more text (please see the whole section 2). Following the other reviewer's suggestion, we rewrote some parts of this section.   

  1. Line 282-285: “Specific structural attributes were developed by pectin's functional groups or attachment of chemicals on the molecule make it a good candidate for several purposes, such as food innovations, nutritional remediation, drug delivery, illness treatment, tissue engineering, and other approaches.” This part suggests that authors cite literature.

 As the referee suggested, references for this paragraph have been added to the new version of the manuscript. The further references are the numbers 99 to 104.

  1. Line 393: “5. Structural models of cross-linked LM pectin and alginates”. In this part, the author enumerates some examples in history and recent years, but does not give the proper differences.

 As the referee suggested, we have added the difference between the different models. In most of the paragraphs of section 5, we have added the difference with the previous model. To see all the changes, please, see section 5 in the new version of our manuscript.

  1. Line 553:“ Properties of pectin that influence the absorption of metal ions” the author can add more discussion in this section.

 We have added more discussion in this section as well as a table including the main technical results, such as the degree of methoxylation and the adsorption capacity (see new table 4) of pectin based-materials, but only considering LM pectin adsorbents.

 Please carefully check the text for writing and grammar problems.

Language mistakes were reviewed in this new version.

Round 2

Reviewer 1 Report

The paper is now suggested for publication. Good luck,